# Almost Optimal Fully Dynamic $k$-Center Clustering with Recourse

**Sayan Bhattacharya** [1]  **Martín Costa** [1]  **Ermiya Farokhnejad** [1]  **Silvio Lattanzi** [2]  **Nikos Parotsidis** [2]

## Abstract

In this paper, we consider the *metric k-center* problem in the fully dynamic setting, where we are given a metric space $(V, d)$ evolving via a sequence of point insertions and deletions and our task is to maintain a subset $S \subseteq V$ of at most $k$ points that minimizes the objective $\max_{x \in V} \min_{y \in S} d(x, y)$. We want to design our algorithm so that we minimize its *approximation ratio*, *recourse* (the number of changes it makes to the solution $S$) and *update time* (the time it takes to handle an update). We give a simple algorithm for dynamic $k$-center that maintains a $O(1)$-approximate solution with $O(1)$ amortized recourse and $\tilde{O}(k)$ amortized update time, *obtaining near-optimal approximation, recourse and update time simultaneously*. We obtain our result by combining a variant of the dynamic $k$-center algorithm of Bateni et al. [SODA'23] with the dynamic sparsifier of Bhattacharya et al. [NeurIPS'23].

## 1. Introduction

Clustering data is a fundamental task in unsupervised learning. In this task, we need to partition the elements of a dataset into different groups (called *clusters*) so that elements in the same group are similar to each other, and elements in different groups are not.

One of the most studied formulations of clustering is *metric k-clustering*, where the data is represented by points in some underlying metric space $(V, d)$, and we want to find a subset $S \subseteq V$ of at most $k$ centers that minimizes some objective function. Due to its simplicity and extensive real-world applications, metric $k$-clustering has been studied extensively for many years and across many computational mod-

els (Charikar et al., 1999; Jain & Vazirani, 2001; Ahmadian et al., 2019; Byrka et al., 2017; Charikar et al., 2003; Ailon et al., 2009; Shindler et al., 2011; Borassi et al., 2020). In this paper, we focus on the *k-center* problem, where the objective function is defined as $\mathrm{cl}(S, V) := \max_{x \in V} d(x, S)$, where $d(x, S) := \min_{y \in S} d(x, y)$. In other words, we want to minimize the maximum distance from any point in $V$ to its nearest point in $S$.

**The dynamic setting:** In recent years, massive and rapidly changing datasets have become increasingly common. If we want to maintain a good solution to a problem defined on such a dataset, it is often not feasible to apply standard computational paradigms and recompute solutions from scratch using *static* algorithms every time the dataset is updated. As a result, to cope with real world-scenarios, significant effort has gone into developing *dynamic* algorithms that are capable of efficiently maintaining solutions as the underlying dataset evolves over time (Bhattacharya et al., 2023b; Behnezhad et al., 2019).

In the case of dynamic clustering, the most practical and studied setting is one where the metric space $(V, d)$ evolves over time via a sequence of *updates* that consist of point insertions and deletions. Dynamic clustering has received a lot of attention from both the theory and machine learning communities over the past decade (Lattanzi & Vassilvitskii, 2017; Chan et al., 2018; Cohen-Addad et al., 2019; Henzinger & Kale, 2020; Bhattacharya et al., 2022; Goranci et al., 2021; Bhattacharya et al., 2024a). This long and influential line of work ultimately aims to design dynamic clustering algorithms with optimal guarantees, mainly focusing on the following three metrics: (I) the *approximation ratio* of the solution $S$ maintained by the algorithm, (II) the *update time* of the algorithm, which is the time it takes for the algorithm to update its solution after a point is inserted or deleted, and (III) the *recourse* of the solution, which is how many points are added or removed from $S$ after an update is performed.[1] These three metrics—approximation ratio, update time, and recourse—capture the essential qualities of a good practical dynamic clustering algorithm: solution quality, efficiency, and stability.

---

[1]Department of Computer Science, University of Warwick [2]Google Research. Correspondence to: Sayan Bhattacharya <s.bhattacharya@warwick.ac.uk>, Martín Costa <martin.costa@warwick.ac.uk>, Ermiya Farokhnejad <ermiya.farokhnejad@warwick.ac.uk>, Silvio Lattanzi <silviol@google.com>, Nikos Parotsidis <nikosp@google.com>.

*Proceedings of the $42^{st}$ International Conference on Machine Learning*, Vancouver, Canada. PMLR 267, 2025. Copyright 2025 by the author(s).

---

[1]In other words, if we let $S$ and $S'$ denote the solution maintained by the algorithm before and after an update, then we define the recourse of the update to be $|S \oplus S'|$, where $\oplus$ denotes symmetric difference.

**The state-of-the-art for dynamic $k$-center:** The classic algorithm of (Gonzalez, 1985) returns a 2-approxmiation to the $k$-center problem in $O(nk)$ time in the static setting, where $n$ is the size of the metric space $V$. It is known to be NP-hard to obtain a $(2 - \epsilon)$-approximation for $\epsilon > 0$ and that any (non-trivial) approximation algorithm for $k$-center has running time $\Omega(nk)$ (Bateni et al., 2023). Thus, the best we can hope for in the dynamic setting is to maintain a 2-approximation in $\tilde{O}(k)$ update time.[2] Bateni et al (Bateni et al., 2023) showed how to maintain a $(2+\epsilon)$-approximation to $k$-center in $\tilde{O}(k/\epsilon)$ update time, obtaining near-optimal approximation and update time. However, their algorithm does not have any non-trivial bound on the recourse (the solution might change completely between updates, leading to a recourse of $\Omega(k)$). In subsequent work, Lacki et al. (Lacki et al., 2024) showed how to maintain a $O(1)$-approximation with $O(1)$ recourse, achieving a worst-case recourse of 4. The algorithm of (Lacki et al., 2024) does however have a large $\tilde{O}(\text{poly}(n))$ update time, with the authors stating that it would be interesting to obtain an update time of $\tilde{O}(\text{poly}(k))$. This was very recently improved upon by (Forster & Skarlatos, 2025), who showed how to obtain an optimal worst-case recourse of 2.[3] Very recently, Bhattacharya et al. (Bhattacharya et al., 2024a) showed how to maintain a $O(\log n \log k)$-approximation with $\tilde{O}(k)$ update time and $\tilde{O}(1)$ recourse, obtaining near-optimal update time and recourse simultaneously, but with polylogarithmic approximation ratio.

Ultimately, *we want to obtain near-optimal approximation, update time and recourse simultaneously*. Each of the algorithms described above falls short in one of these metrics, having either poor approximation, update time or recourse. This leads us to the following natural question:

> **Q: Can we design a dynamic $k$-center algorithm with $O(1)$-approximation, $\tilde{O}(k)$ update time and $O(1)$ recourse?**

**Our contribution:** We give an algorithm for dynamic $k$-center that answers this question in the affirmative, obtaining the following result.

**Theorem 1.1** (informal)**.** *There is an algorithm for dynamic $k$-center that maintains a 20-approximation with $O(k \log^5(n) \log \Delta)$ update time and $O(1)$ recourse.*

We emphasise that our algorithm is *significantly simpler* than the state-of-the-art dynamic $k$-center algorithms described above. Our starting point is the algorithm of (Bateni

et al., 2023). Even though the high-level framework used by (Bateni et al., 2023) is quite simple, they require a significant amount of technical work to obtain good update time; we bypass this by using the dynamic sparsifier of (Bhattacharya et al., 2023a), giving a very simple variant of their algorithm that has good recourse and can be implemented efficiently using the dynamic MIS algorithm of (Behnezhad et al., 2019) as a black box.[4]

**Related work:** It was first shown how to maintain a $(2+\epsilon)$-approximation to $k$-center with $\tilde{O}(k^2)$ update time by (Chan et al., 2018); the algorithm of (Chan et al., 2022) improved the space efficiency of this algorithm, but at the cost of obtaining a $(4 + \epsilon)$-approximation. Dynamic $k$-clustering has also been explored in various other settings; there are lines of work that consider the specific settings of Euclidean spaces (Bateni et al., 2023; Bhattacharya et al., 2024b), other $k$-clustering objectives such as $k$-median, $k$-means and facility location (Cohen-Addad et al., 2019; Henzinger & Kale, 2020; Bhattacharya et al., 2023a; 2024a; 2022; 2025), additional constraints such as *outliers* (Biabani et al., 2023) and the incremental (insertion only) setting, where the objective is to minimize recourse while ignoring update time (Lattanzi & Vassilvitskii, 2017; Fichtenberger et al., 2021). A separate line of work considers metric spaces of *bounded doubling dimension*, where it is known how to maintain a $(2+\epsilon)$-approximation to $k$-center in $\tilde{O}(1)$ update time (Goranci et al., 2021; Gan & Golin, 2024; Pellizzoni et al., 2025). (Cruciani et al., 2024) also considered the problem of dynamic $k$-center w.r.t. the shortest path metric on a graph undergoing edge insertions and deletions.

### 1.1. Technical Overview

We obtain our result by combining a simple variant of the algorithm of (Bateni et al., 2023) with the dynamic sparsification algorithm of (Bhattacharya et al., 2023a). In this technical overview, we give an informal description of a variant of the algorithm of (Bateni et al., 2023), and then explain how it can be modified to obtain our result.

**The algorithm of (Bateni et al., 2023):** The dynamic algorithm of (Bateni et al., 2023) works by leveraging the well-known reduction from $k$-center to *maximal independent set* (MIS) using *threshold graphs* (see Definition 1.6) (Hochbaum & Shmoys, 1986).

The algorithm maintains a collection of $\tilde{O}(1)$ many threshold graphs $\{G_{\lambda_i}\}_i$ of $(V, d)$ and MISs $\{\mathcal{I}_i\}_i$ of these threshold graphs for values $\lambda_i$ that increase in powers of $(1 + \epsilon)$,

---

[2]The $\tilde{O}(\cdot)$ notation hides polylogarithmic factors in $n$ and the *aspect ratio* of the metric space $\Delta$ (see Section 1.2).

[3]We note that (Lacki et al., 2024; Forster & Skarlatos, 2025) define the recourse of an update as the number of centers that are *swapped* assuming that the size of the solution is always $k$. Thus, the bounds in their papers are smaller by a factor of 2.

[4]For ease of exposition, we avoid discussing certain details of the algorithms in the introduction, such as oblivious vs. adaptive adversaries, amortized vs. worst-case guarantees, and so on. Table 1 in Appendix A contains a more comprehensive summary of the state-of-the-art algorithms for fully dynamic $k$-center in general metric spaces.

i.e. $\lambda_i = (1 + \epsilon) \cdot \lambda_{i-1}$. The MISs $\{\mathcal{I}_i\}_i$ satisfy the following property: *Let $i^\star$ be the smallest index such that $|\mathcal{I}_{i^\star}| \leq k$, then $\mathcal{I}_{i^\star}$ is a $(2 + O(\epsilon))$-approximation to the $k$-center problem on $(V, d)$.* At any point in time, the output of this algorithm is the MIS $\mathcal{I}_{i^\star}$.

**Implementation:** Using the dynamic MIS algorithm of Behnezhad et al. (2019) as a black box, which can maintain an MIS under vertex insertions and deletions in $\tilde{O}(n)$ update time, we can implement this algorithm with $\tilde{O}(n)$ update time. We note that this algorithm is a simple variant of the actual algorithm presented in (Bateni et al., 2023), which is much more technical. Since they want an update time of $\tilde{O}(k)$, it is not sufficient to use the algorithm of (Behnezhad et al., 2019) as a black box and instead need to use the internal data structures and analysis intricately. We later improve the update time to $\tilde{O}(k)$ using a completely different approach.

**Obtaining good recourse:** It's known how to maintain a *stable* MIS of a dynamic graph, so that vertex updates in the graph lead to few changes in the MIS (Behnezhad et al., 2019). However, the main challenge that we encounter while trying to obtain good recourse is that the MISs $\{\mathcal{I}_i\}_i$ do not necessarily have any relation to each other. Thus, whenever the index $i^\star$ changes, the recourse could be as bad as $\Omega(k)$. By modifying the algorithm, we can ensure that the MISs $\{\mathcal{I}_i\}_i$ are *nested*, and thus switching between them does not lead to high recourse. This leads to the following theorem.

**Theorem 1.2.** *There is an algorithm for dynamic $k$-center against oblivious adversaries that maintains an $8$-approximation with $O(n \log^4(n) \log \Delta)$ expected worst-case update time and $4$ expected worst-case recourse.*

To improve the update time from $\tilde{O}(n)$ to $\tilde{O}(k)$, we use the dynamic sparsifier of (Bhattacharya et al., 2023a). This allows us to assume that the underlying metric space has size $\tilde{O}(k)$, thus leading to an update time of $\tilde{O}(k)$. This leads to the following theorem.

**Theorem 1.3.** *There is an algorithm for dynamic $k$-center against oblivious adversaries that maintains a $20$-approximation with $O(k \log^5(n) \log \Delta)$ expected amortized update time and $O(1)$ expected amortized recourse.*[5]

*Remark* 1.4. In Appendix C, we design a different sparsifier by building on top of the sparsifier of (Bhattacharya et al., 2023a), allowing us to obtain a recourse of at most $8 + \epsilon$.

### 1.2. Preliminaries and Notations

We now provide definitions and notations that we use throughout the paper.

**Definition 1.5** (Metric Space). Consider a set of points

---

[5]Here, the approximation guarantee holds w.h.p.

$V$ and a distance function $d : V \times V \to \mathbb{R}^{\geq 0}$ such that $d(x, x) = 0$, $d(x, y) = d(y, x)$ and $d(x, y) \leq d(x, z) + d(z, y)$ for all $x, y, z \in V$. We refer to the pair $(V, d)$ as a metric space.

The *aspect ratio* of $(V, d)$ is defined as the ratio between the maximum and minimum non-zero distance of any two points in the space. For each $S \subseteq V$ and $x \in V$, we denote the distance from $x$ to $S$ by $d(x, S) := \min_{y \in S} d(x, y)$. The *ball* around $x \in V$ of radius $r$ is defined as $B(x, r) := \{y \in V \mid d(x, y) \leq r\}$.

**Metric $k$-center:** In the metric $k$-center problem, we are given a metric space $(V, d)$ and the objective is to find $S \subseteq V$ of size at most $k$ that minimizes $\mathrm{cl}(S, V) := \max_{x \in V} d(x, S)$. We denote the cost of the optimum solution to the $k$-center problem by

$$\mathrm{OPT}_k(V) := \min_{S \subseteq V, |S| \leq k} \mathrm{cl}(S, V).$$

We sometimes abbreviate $\mathrm{cl}(S, V)$ by $\mathrm{cl}(S)$ and $\mathrm{OPT}_k(V)$ by $\mathrm{OPT}_k$ when the set $V$ is clear from the context.

**Definition 1.6** ($\lambda$-Threshold Graph). Given a metric space $(V, d)$ and any $\lambda \geq 0$, we define the $\lambda$-*threshold graph* of $(V, d)$ to be the graph $G_\lambda := (V, E_\lambda)$ where $E_\lambda := \{(x, y) \in \binom{V}{2} \mid d(x, y) \leq \lambda\}$.

**Definition 1.7** (Bicriteria Approximation). Given a $\rho$-metric space $(V, d)$, we say that a subset of points $U \subseteq V$ is an $(\alpha, \beta)$-*approximation* to the $(k, p)$-clustering problem if $\mathrm{cl}_p(U) \leq \alpha \cdot \mathrm{OPT}_k(V)$ and $|U| \leq \beta k$.

**Definition 1.8** ($(\alpha, \beta)$-Sparsifier). Fix a $\rho \geq 1$ and a clustering problem. An $(\alpha, \beta)$-*sparsifier* is a dynamic algorithm that, given a $\rho$-metric space $(V, d)$ undergoing point insertions and deletions, maintains $U \subseteq V$ which is an $(\alpha, \beta)$-approximation for the clustering problem at any point in time. The recourse of the sparsifier is defined as the number of insertions and deletions of the points in the maintained solution $U$.

For any natural number $N$, we denote the set $\{1, 2, \ldots, N\}$ by $[N]$. Consider a sequence of updates $\sigma_1, \ldots, \sigma_T$. For each point set $U$ that is maintained explicitly by our algorithm, we use $U^{(t)}$ (for any $t \in [T]$) to indicate the status of this set after handling update $\sigma_t$. For instance, if $S$ is the solution maintained by our algorithm, $S^{(t)}$ is the output after handling update $\sigma_t$. We also define $\delta_t(U) := U^{(t-1)} \oplus U^{(t)}$ to indicate the symmetric difference of the maintained set $U$ before and after handling update $\sigma_t$.

## 2. Our Dynamic Algorithm (Theorem 1.2)

In this section, we provide our dynamic $k$-center algorithm for Theorem 1.2, which we call `Dynamic-k-Center`. We begin by describing a dynamic algorithm for *maximal*

*independent set* (MIS) that our algorithm uses as a black box.

### 2.1. The Algorithm `DynamicMIS`

(Behnezhad et al., 2019) present an algorithm, which we refer to as `DynamicMIS`, that, given a dynamic graph $G$ undergoing updates via a sequence of *node insertions and deletions*, explicitly maintains an MIS $\mathcal{I}$ of $G$. The following lemma summarizes the key properties of `DynamicMIS`.

**Lemma 2.1.** *The algorithm* `DynamicMIS` *has an expected worst-case update time of $O(n \log^4 n)$ and an expected worst-case recourse of $1$.*[6]

### 2.2. Our Algorithm: `Dynamic-k-Center`

Let $(V, d)$ be a dynamic metric space undergoing updates via a sequence of point insertions and deletions, and let $d_{\max}$ and $d_{\min}$ be upper and lower bounds on the maximum and minimum (non-zero) distance between any two points in the space at any time throughout the sequence of updates. Let $\lambda_i := 2^{i-2} \cdot d_{\min}$ and $G_i := G_{\lambda_i}$ for each $0 \leq i \leq \tau$, where $\tau := \log_2 \Delta + 2$ and $\Delta$ is the aspect ratio (see Section 1.2) of the underlying metric space. Note that the edge-sets of these threshold graphs are nested, i.e., $E_1 \subseteq \cdots \subseteq E_\tau$. Let $\mathcal{I}_0$ denote the set $V$. Our algorithm `Dynamic-k-Center` maintains the following for each $i \in [\tau]$:

- An MIS $\mathcal{I}_i := $ `DynamicMIS`$(G_i[\mathcal{I}_{i-1}])$ of $G_i[\mathcal{I}_{i-1}]$, where $G[S]$ denotes the subgraph of $G$ induced by the node-set $S$.

- The set $\mathcal{I}_{i-1} \setminus \mathcal{I}_i$ ordered lexicographically.

Let $i^\star \in [\tau]$ be the smallest index such that $|\mathcal{I}_{i^\star}| \leq k$. Then the **output** $S$ of our dynamic algorithm is the union of the set $\mathcal{I}_{i^\star}$ and the first $k - |\mathcal{I}_{i^\star}|$ points in the set $\mathcal{I}_{i^\star-1} \setminus \mathcal{I}_{i^\star}$.

### 2.3. Analysis of Our Algorithm

We begin by bounding the approximation ratio of our algorithm and then proceed to analyze the recourse and update time.

**Approximation ratio:** We now show that the set $\mathcal{I}_{i^\star}$ is an 8 approximation to the $k$-center problem on $(V, d)$. Since $S \supseteq \mathcal{I}_{i^\star}$, the approximation guarantee of our algorithm follows. We begin with the following simple lemmas.

**Lemma 2.2.** *For each $i \in [\tau]$, we have that $\mathrm{cl}(\mathcal{I}_i) \leq 2\lambda_i$.*

*Proof.* We prove this by induction on $i$. We first note that, since $\mathcal{I}_0 = V$, $\mathrm{cl}(\mathcal{I}_0) = 0$. Now, let $i \in [\tau]$ and $x \notin \mathcal{I}_i$, and assume that the lemma holds for $i - 1$. Since $\mathrm{cl}(\mathcal{I}_{i-1}) \leq$

---

$2\lambda_{i-1}$, there is some $y \in \mathcal{I}_{i-1}$ such that $d(x, y) \leq 2\lambda_{i-1}$. If $y \in \mathcal{I}_i$, then $d(x, \mathcal{I}_i) \leq 2\lambda_{i-1} = \lambda_i$ and we are done. Otherwise, since $y \in \mathcal{I}_{i-1} \setminus \mathcal{I}_i$ and $\mathcal{I}_i$ is an MIS, there exists some $z \in \mathcal{I}_i$ such that $d(y, z) \leq \lambda_i$. Thus, we have that $d(x, \mathcal{I}_i) \leq d(x, z) \leq d(x, y) + d(y, z) \leq 2\lambda_{i-1} + \lambda_i = 2\lambda_i$. $\square$

**Lemma 2.3.** *For each $i \in [\tau]$ such that $|\mathcal{I}_i| > k$, we have that $\lambda_i \leq 2 \cdot \mathrm{OPT}_k$.*

*Proof.* Let $S^\star$ denote an optimal solution to the $k$-center problem in $(V, d)$ and let $B^\star := B(S^\star, \lambda_i/2) = \cup_{y^\star \in S^\star} B(y^\star, \lambda_i/2)$ (see Section 1.2 for the definition of the ball $B(x, r)$). Given any point $y^\star \in S^\star$, and any two distinct points $y, y' \in \mathcal{I}_i$, we can see that at most one of $y$ and $y'$ is contained in $B(y^\star, \lambda_i/2)$, otherwise $d(y, y') \leq d(y, y^\star) + d(y^\star, y') \leq \lambda_i$, contradicting the fact that $\mathcal{I}_i$ is an MIS since $(y, y') \in E_{\lambda_i}$. Combining this with our assumption that $|\mathcal{I}_i| > k \geq |S^\star|$, it follows that $\mathcal{I}_i \setminus B(S^\star, \lambda_i/2) \neq \emptyset$, as otherwise by pigeonhole principle at least two different elements of $\mathcal{I}_i$ would be contained in the same $B(y^\star, \lambda_i/2)$ for some $y^\star \in S^\star$, which is in contradiction with the above explanation. Hence $\mathrm{cl}(S^\star) \geq \lambda_i/2$. It follows that $\lambda_i \leq 2 \cdot \mathrm{OPT}_k$. $\square$

Applying Lemmas 2.2 and 2.3 and noting that $|\mathcal{I}_{i^\star-1}| > k$, we get $\mathrm{cl}(\mathcal{I}_{i^\star}) \leq 2\lambda_{i^\star} = 4\lambda_{i^\star-1} \leq 8 \cdot \mathrm{OPT}_k$.

**Recourse:** We now proceed to bound the expected recourse of our algorithm. Suppose that our algorithm handles a sequence of updates $\sigma_1, \ldots, \sigma_T$. Consider $\mathcal{I}_i^{(t)}, \delta_t(\mathcal{I}_i), S^{(t)}$ $\delta_t(S)$ for each $t \in [T]$ (recall the notation from Section 1.2).

The following lemma shows that the expected recourse of each $\mathcal{I}_i$ is small.

**Lemma 2.4.** *For each $t \in [T]$, $i \in [\tau]$, we have that $\mathbb{E}[|\delta_t(\mathcal{I}_i)|] \leq 1$.*

*Proof.* Fix any $t \in [T]$. We prove this by induction on $i$. We first note that $\mathbb{E}[|\delta_t(\mathcal{I}_0)|] = |\delta_t(V)| = 1$. Now, let $i \in [\tau]$ and assume that the lemma holds for $i - 1$. Then $\mathbb{E}[|\delta_t(\mathcal{I}_i)|]$ is the expected recourse of the solution maintained by the dynamic algorithm `DynamicMIS`$(G_i[\mathcal{I}_{i-1}])$. By Lemma 2.1, each node update in $G_i[\mathcal{I}_{i-1}]$ leads to an expected recourse of at most 1 in the solution maintained by this algorithm. Hence, we have that $\mathbb{E}[|\delta_t(\mathcal{I}_i)|] \leq |\delta_t(\mathcal{I}_{i-1})|$. Taking expectation on both sides, the lemma follows. $\square$

We now use Lemma 2.4 to bound the expected recourse of the solution $S$.

**Lemma 2.5.** *For each $t \in [T]$, $\mathbb{E}[|\delta_t(S)|] \leq 4$.*

*Proof.* Let $j$ denote the value of the index $i^\star$ immediately after handling the update $\sigma_{t-1}$. If $j = 0$, we have $S^{(t-1)} =$

$\mathcal{I}_0^{(t-1)}$ which equals the whole space at this time whose size is at most $k$. In this case, it is obvious that $|\delta_t(S)| = |S^{(t)} \oplus S^{(t-1)}| \leq 1$. If $j > 0$, the size of the whole space, after updates $t-1$ and $t$, is at least $k$.

*Claim* 2.6. We have

$$|S^{(t-1)} \oplus S^{(t)}| \leq 2|\delta_t(\mathcal{I}_j)| + 2|\delta_t(\mathcal{I}_{j-1})|.$$

*Proof.* Consider the following ordering on the points of $V$.

$$\mathcal{I}_\tau, \mathcal{I}_{\tau-1} \setminus \mathcal{I}_\tau, \mathcal{I}_{\tau-2} \setminus \mathcal{I}_{\tau-1}, \cdots, \mathcal{I}_0 \setminus \mathcal{I}_1,$$

where for each $i \in [\tau]$, elements of $\mathcal{I}_{i-1} \setminus \mathcal{I}_i$ are written in the lexicographic order. The output of the algorithm is exactly the first $k$ points in this order. We assumed that $S^{(t-1)}$ consists of $\mathcal{I}_j^{(t-1)}$ and the first $k - |\mathcal{I}_j^{(t-1)}|$ elements of $\mathcal{I}_{j-1}^{(t-1)} \setminus \mathcal{I}_j^{(t-1)}$. As a result, after handling update $\sigma_t$, we have at most $|\delta_t(\mathcal{I}_j)| + |\delta_t(\mathcal{I}_{j-1})|$ many changes in the first $k$ points in this order. Hence, there are at most these many deletions from $S^{(t-1)}$ as well as these many insertions, which concludes the claim. $\square$

According to the above claim, by taking expectations and applying Lemma 2.4, we have that $\mathbb{E}[|\delta_t(S)|] \leq 4$. $\square$

**Update time:** Fix any $t \in [T]$ and $i \in [\tau]$. If we delete $x$ from $\mathcal{I}_i$, then we remove the corresponding node from $G[\mathcal{I}_i]$. By Lemma 2.1, the expected time to update the MIS $\mathcal{I}_{i+1}$ in this graph after the deletion of $x$ is $\tilde{O}(n)$. If we insert a new point $x$ into $\mathcal{I}_i$, we insert a node $x$ into $G[\mathcal{I}_i]$. To find the edges between $x$ and other nodes, we first find the distance of $x$ to all other points in $\mathcal{I}_i$ in $O(|\mathcal{I}_i|) = O(n)$ time and then compare the distance with $\lambda_i$. Again, by Lemma 2.1, the expected time to update the MIS $\mathcal{I}_{i+1}$ in this graph after inserting the node $x$ is $O(n \log^4 n)$.

Hence, we can perform both insertions into and deletions from $\mathcal{I}_i$ in time $O(n \log^4 n)$ for each $i \in [\tau]$. By Lemma 2.4, the expected number of updates in $\mathcal{I}_i$ is at most 1. Thus, we can update the graph $G[\mathcal{I}_i]$ and $\mathcal{I}_{i+1}$ in expected $O(n \log^4 n)$ time. Finally, since $\tau = O(\log \Delta)$, the total time taken to update all of the graphs $G[\mathcal{I}_i]$ and MISs $\mathcal{I}_i$ with $O(n \log^4(n) \log \Delta)$ in expectation.

In order to maintain $S$, for each $i$, we can maintain two binary search trees $\mathcal{T}_i^f$ and $\mathcal{T}_i^r$ such that $\mathcal{T}_i^f$ contains the first $k - |\mathcal{I}_i|$ elements of $\mathcal{I}_{i-1} \setminus \mathcal{I}_i$ and $\mathcal{T}_i^r$ contains the rest of elements in $\mathcal{I}_{i-1} \setminus \mathcal{I}_i$. After each insertion or deletion in any of $\mathcal{I}_i$ or $\mathcal{I}_{i-1}$, we can exchange the last element of $\mathcal{T}_i^f$ and the first element of $\mathcal{T}_i^r$ appropriately, in order to maintain the property in the definition of $\mathcal{T}_i^f$ and $\mathcal{T}_i^r$. For instance after a deletion in $\mathcal{I}_i$, the value of $k - |\mathcal{I}_i|$ increments, and we should remove the first element of $\mathcal{T}_i^r$ and add it to $\mathcal{T}_i^f$ (which would be new last element). Hence, we have the first

$k - |\mathcal{I}_i|$ elements of $\mathcal{I}_{i-1} \setminus \mathcal{I}_i$ stored in $\mathcal{T}_i^f$ explicitly, at any time. Note that

$$\mathbb{E}[|\delta_t(\mathcal{I}_{i-1} \setminus \mathcal{I}_i)|] \leq \mathbb{E}[|\delta_t(\mathcal{I}_{i-1})|] + \mathbb{E}[|\delta_t(\mathcal{I}_i)|] \leq 2,$$

where the last inequality follows from Lemma 2.4. This means that the update time for maintaining each of these trees is $O(\log n)$ in expectation.

In total, we can maintain all of the objects in our algorithm in $O(n \log^4(n) \log \Delta)$ update time in expectation.

# 3. Improving the Update Time (Theorem 1.3)

We now show how to use *sparsification* to improve the update time of our algorithm to $\tilde{O}(k)$. In Section 3.1, we show how to use a sparsifier (see Definition 1.8) as a black box in order to speed up a dynamic algorithm. In Section 3.2, we describe the guarantees of a sparsifier for $k$-center and combine it with Dynamic-$k$-Center, proving Theorem 1.3. The rest of the Section 3 is devoted to constructing this sparsifier for $k$-center, which follows from the previous work of (Bhattacharya et al., 2023a). Since their algorithm is primarily designed for $k$-median rather than $k$-center, we describe the sparsifier of (Bhattacharya et al., 2023a) in Section 3.3, along with a new analysis that gives an improved bound on its approximation ratio for $k$-center and its recourse.

## 3.1. Dynamic Sparsification

The following theorem describes the properties of the dynamic algorithm obtained by *composing* a sparsifier and any dynamic algorithm for $k$-center problem.

**Theorem 3.1.** *Assume we have an $(\alpha_S, \beta)$-sparsifier for metric $k$-center with $T_S$ update time and $R_S$ recourse, and a dynamic $\alpha_A$-approximation algorithm for metric $k$-center with $T_A(n)$ update time and $R_A(n)$ recourse. Then we can obtain a dynamic algorithm for metric $k$-center with $(\alpha_S + 2\alpha_A)$-approximation ratio, $O(T_S + R_S \cdot T_A(\beta k))$ update time and $O(R_S \cdot R_A(\beta k))$ recourse.*[7]

*Proof.* Let $(V, d)$ be a dynamic metric space. We run the sparsifier on this space which at any point in time maintains a subset $U \subseteq V$. This defines a dynamic metric subspace $(U, d)$ of size at most $\beta k$, such that each update in $V$ leads to at most $R_S$ updates in $U$ and $\text{cl}(U, V) \leq \alpha_S \cdot \text{OPT}_k(V)$.

Now, we feed the new dynamic subspace $(U, d)$ (whose size is at most $\beta k$ at any time) to the $k$-center algorithm, which maintains a subset of points $S \subseteq U$ of size $k$ such that $\text{cl}(S, U) \leq \alpha_A \cdot \text{OPT}_k(U)$. We refer to this process as *composing* these dynamic algorithms.

Since each update in $V$ leads to at most $R_S$ updates in $U$, it follows that the update time and recourse of the composite

---

[7]This holds for both worst-case and amortized guarantees.

algorithm are $O(T_S + R_S \cdot T_A(\beta k))$ and $O(R_S \cdot R_A(\beta k))$ respectively. To bound the approximation ratio, consider the following claim.

*Claim* 3.2. Given subsets $S \subseteq U \subseteq V$, we have that $\text{cl}(S, V) \leq \text{cl}(U, V) + \text{cl}(S, U)$.

*Proof.* Let $x \in V$, $y$ and $y'$ be the closest points to $x$ in $S$ and $U$ respectively, and $y^\star$ be the closest point to $y'$ in $S$. Then we have that

$$d(x, S) = d(x, y) \leq d(x, y^\star) \leq d(x, y') + d(y', y^\star)$$
$$= d(x, U) + d(y', S) \leq \text{cl}(U, V) + \text{cl}(S, U).$$

It follows that $\text{cl}(S, V) \leq \text{cl}(U, V) + \text{cl}(S, U)$. $\qquad\square$

Applying Claim 3.2, we get that

$$\text{cl}(S, V) \leq \text{cl}(U, V) + \text{cl}(S, U)$$
$$\leq \alpha_S \cdot \text{OPT}_k + \alpha_A \cdot \text{OPT}_k(U)$$
$$\leq (\alpha_S + 2\alpha_A) \cdot \text{OPT}_k(V),$$

where the last inequality follows since $\text{OPT}_k(W) \leq 2 \cdot \text{OPT}_k(V)$ for any $W \subseteq V$ (see Lemma B.1). $\qquad\square$

### 3.2. Proof of Theorem 1.3

We start with the following lemma, which provides the guarantees of a sparsifier for the $k$-center problem. We prove this lemma in Section 3.3.

**Lemma 3.3.** *There exists a $(4, O(\log(n/k)))$-sparsifier for the $k$-center problem on a metric space $(V, d)$, whose approximation guarantee holds with high probability, and has $O(k \log(n/k))$ amortized update time and $O(1)$ amortized recourse.*[8]

Theorem 1.3 now follows by combining our algorithm `Dynamic-k-Center` from Theorem 1.2 and the sparsifier of Lemma 3.3 by using the composition described in Theorem 3.1. The approximation ratio of the resulting algorithm is $\alpha_S + 2\alpha_A = 4 + 2 \cdot 8 = 20$ w.h.p. The expected amortized recourse is $O(R_S \cdot R_A(\beta k)) = O(1)$ since both $R_S$ and $R_A$ are constant. For the update time, we have $T_S = O(k \log(n/k))$, $R_S = 1$, $\beta = \log(n/k)$, and $T_A(\beta k) = O((\beta k) \log^4(\beta k) \log \Delta) = O(k \log^5(n) \log \Delta)$. Hence, the final update time of the algorithm is $O(T_S + R_S \cdot T_A(\beta k)) = O(k \log^5(n) \log \Delta)$.

### 3.3. The Algorithm `Sparsifier` (Proof of Lemma 3.3)

The algorithm in Lemma 3.3 (which we refer to as `Sparsifier`) follows from the results of (Bhattacharya

et al., 2023a). Before we provide the algorithm, we mention that in (Bhattacharya et al., 2023a), the main goal is to solve the $k$-median problem. In particular, there are some technical details for $k$-median that we do not need for $k$-center. There is no argument for recourse of the algorithm in (Bhattacharya et al., 2023a). Here, we briefly describe the algorithm and show that the amortized recourse of this algorithm is actually $O(1)$.

The algorithm `Sparsifier` is a dynamization of a well-known algorithm by Mettu and Plaxton (Mettu & Plaxton, 2002).

**The algorithm of Mettu-Plaxton:** The main component of this algorithm is the following key subroutine:

- `AlmostCover(U)`: Sample a subset $S \subseteq U$ of size $2k$ u.a.r., compute the subset $C \subseteq U$ of the $|U|/4$ points in $U$ that are closest to the points in $S$, and return $(S, U \setminus C)$.

Intuitively, this subroutine grows balls centered at sampled points simultaneously until they cover a quarter of the space. This gives a partitioning of a quarter of the points into $O(k)$ sets, which we refer to as *clusters*.

The algorithm begins by setting $U_1 := V, i = 0$, and, while $|U_i| \geq \Theta(k)$, repeatedly sets $(S_i, U_{i+1}) \leftarrow$ `AlmostCover(U_i)`. This defines a sequence of nested subsets $V = U_1 \supseteq \cdots \supseteq U_\ell$ for $\ell = O(\log(n/k))$ and subsets $S_i \subseteq U_i$ for each $i \in [\ell - 1]$. The output of the algorithm is the set $U := S_1 \cup \cdots \cup S_{\ell-1} \cup U_\ell$, which has size at most $O(k\ell) \leq O(k \cdot \log(n/k))$. Note that for each $i$, each point $x \in S_i$ is the center of a cluster and points of $U_\ell$ are singleton clusters. Hence, the final clustering produced by the algorithm assigns each point $x \in U_i \setminus U_{i+1}$ to its nearest center in $S_i$ (breaking the ties arbitrarily).

**The algorithm `Sparsifier`:** At a high level, (Bhattacharya et al., 2023a) dynamizes the static algorithm of Mettu-Plaxton by maintaining an approximate version of their hierarchy of nested sets, which are updated lazily and periodically reconstructed by using `AlmostCover` in the same way as the Mettu-Plaxton algorithm.[9] More specifically, `Sparsifier` maintains the following

- Subsets $V = U_1 \supseteq \cdots \supseteq U_\ell$ for $\ell = O(\log(n/k))$.
- $S_i \subseteq U_i$ for each $i \in [\ell - 1]$.
- $U := S_1 \cup \cdots \cup S_{\ell-1} \cup U_\ell$ as the output.

We now describe how these items are updated as points are inserted and deleted from $V$.

---

[8]In Appendix C, we show how to construct a different sparsifier, leading to a recourse of at most $8 + \epsilon$.

[9]Here we slightly change the algorithm as follows. We will call `AlmostCover` multiple times instead of once, and then select the best output among these independent calls in order to boost the approximation ratio (see Lines 9 to 13 in Algorithm 1).

*Insertion:* When a point $x$ is inserted into $V$, $x$ is added to each set in $U_1, \ldots, U_\ell$. In this case, $x$ would be a new singleton cluster.

*Deletion:* When a point $x$ is deleted from $V$, $x$ is removed from each set in $U_1, \ldots, U_\ell$. If $x$ is contained in some $S_i$, then $x$ is removed from $S_i$ and replaced with any other point currently in its cluster as the new center (if its cluster is non-empty).

We refer to the above updates as *lazy* updates. After performing a lazy update, we have the reconstruction phase as follows.

*Reconstruction:* For each $i \in [\ell]$, the algorithm periodically reconstructs *layer $i$* and all subsequent layers (i.e. the sets $S_i, \ldots, S_\ell$ and $U_{i+1}, \ldots, U_\ell$) every $\Omega(|U_i|)$ updates. To do this, for each $i \in [\ell]$, we keep track of the number of updates on $U_i$ after the last time it was reconstructed. After each insertion and deletion, we find the smallest index $j$ where the number of updates on $U_{j+1}$ since the last time it was reconstructed is at least $|U_j|/4$ (note that $|U_j|$ also varies over time). See Algorithm 1 for more details. The reconstruction is done using AlmostCover in the same way as the static algorithm described above, essentially running the Mettu-Plaxton algorithm starting with the input $U_j$.[10] We note that, after a reconstruction, the value of $\ell$ can change.

---

**Algorithm 1** Reconstruct

1: **for** $i = 1$ **to** $\ell$ **do**
2:    COUNT$[i]$ = COUNT$[i]$ + 1
3: **end for**
4: $j$ = smallest index s.t. COUNT$[j] \geq |U_j|/4$
5: **for** $i = j$ **to** $\ell$ **do**
6:    COUNT$[i]$ = 0
7: **end for**
8: **repeat**
9:    **for** $m = 1$ **to** $M = \Theta(\log n)$ **do**
10:       $(A_m, B_m) \leftarrow$ AlmostCover$(U_j)$
11:    **end for**
12:    $m^\star \leftarrow \arg\min_{1 \leq m \leq M} \text{cl}(A_m, U_j \setminus B_m)$
13:    $(S_j, U_{j+1}) \leftarrow (A_{m^\star}, B_{m^\star})$
14:    $j = j + 1$
15: **until** $|U_j| \leq 16k$
16: $\ell = j$
17: $U := S_1 \cup \cdots \cup S_{\ell-1} \cup U_\ell$

---

### 3.3.1. APPROXIMATION RATIO ANALYSIS

We show the solution $U$ maintained by the Sparsifier is $(4, O(\log(n/k)))$-approximate.

**Lemma 3.4** (Lemma 3.2, (Bhattacharya et al., 2023a))**.** *We*

have $\ell = O(\log(n/k))$ *at any point in time during the execution of* Sparsifier.

This lemma implies that the size of the solution $U \subseteq V$ maintained by the algorithm is always $\sum_{i=0}^{\ell-1} |S_i| + |U_\ell| = \ell \cdot O(k) = O(k \cdot \log(n/k))$.

**Lemma 3.5.** *The solution $U$ maintained by the algorithm* Sparsifier *is $4$-approximate w.h.p.*

According to Lemma 3.4 and Lemma 3.5, we can see that the solution $U$ is a $(4, O(\log(n/k)))$-approximation as desired. Now, we proceed with the proof of Lemma 3.5.

*Proof of Lemma 3.5.* For every $C \subseteq V$, define

$$\text{diam}(C) := \max_{x,y \in C,\ x \neq y} d(x, y).^{11}$$

For each $W \subseteq V$ and $0 < \beta \leq 1$, define $\mu_k^\beta(W)$ to be the minimum real number $\mu > 0$ such that there exist $k$ disjoint subsets $C_1, C_2, \cdots, C_k \subseteq W$, such that

$$\sum_{i \in [k]} |C_i| \geq \beta \cdot |W| \text{ and } \forall i \in [k], \text{diam}(C_i) \leq \mu.$$

We start with a series of claims.

*Claim* 3.6. We have $\mu_k^1(V) \leq 2 \cdot \text{OPT}_k(V)$.

*Proof.* Consider an optimal solution $S^\star \subseteq V$ for $k$-center on $V$ and the clusters $C_1, C_2, \ldots, C_k$ corresponding to the centers $s_1, s_2, \ldots, s_k$ in $S^\star$. For each $i \in [k]$ and each $x, y \in C_i$, we have $d(x, y) \leq d(x, s_i) + d(s_i, y) \leq 2 \cdot \text{OPT}_k(V)$. Hence, $\text{diam}(C_i) \leq 2 \cdot \text{OPT}_k(V)$ for each $i \in [k]$. Since $C_1, \ldots, C_k$ is a partition of $V$, all the conditions in the definition of $\mu_k^1(V)$ are satisfied, which concludes $\mu_k^1(V) \leq 2 \cdot \text{OPT}_k(V)$. $\square$

*Claim* 3.7. For every $W \subseteq V$, we have $\mu_k^1(W) \leq \mu_k^1(V)$.

*Proof.* For every collection of subsets $\{C_i\}_{i \in [k]}$ of $V$ satisfying the conditions in the definition of $\mu_k^1(V)$, the collection $\{C_i \cap W\}_{i \in [k]}$ satisfy the same conditions in the definition of $\mu_k^1(W)$. Hence, $\mu_k^1(W) \leq \mu_k^1(V)$. $\square$

*Claim* 3.8. For every $W$ and $W'$ satisfying $|W \oplus W'| \leq |W|/4$, we have $\mu_k^{1/2}(W') \leq \mu_k^1(W)$.

*Proof.* Let $\mu^\star = \mu_k^1(W)$ and assume $\{C_i\}_{i \in [k]}$ is the optimal collection in the definition of $\mu_k^1(W)$ such that $\text{diam}(C_i) \leq \mu^\star$ for all $i \in [k]$. Define $C_i' := C_i \cap W'$ for each $i \in [k]$. We show that $\{C_i'\}_{i \in [k]}$ satisfy the conditions in the definition of $\mu_k^{1/2}(W')$. Obviously, $\text{diam}(C_i') = \text{diam}(C_i \cap W') \leq \text{diam}(C_i) \leq \mu^\star$. Since $\{C_i\}_{i \in [k]}$ is a partitioning of $W$, we have $|\cup_{i \in [k]} C_i'| = |(\cup_{i \in [k]} C_i) \cap W'| =$

---

[10]For example, after reconstructing layer 1, the outputs of the dynamic algorithm and the Mettu-Plaxon algorithm are the same.

[11]If $|C| = 1$, simply define $\text{diam}(C) = 0$.

$|W \cap W'| \geq |W'|/2$. The last inequality follows from $|W \oplus W'| \leq |W|/4$ and a simple counting argument. $\qquad \square$

*Claim* 3.9. Consider a single call to $\texttt{AlmostCover}(W)$, where $S \subseteq W$ is sampled and the balls around $S$ of radius $r$ cover $C$ such that $C = |W|/4$. Then, with constant probability, we have $r \leq \mu_k^{1/2}(W)$.

*Proof.* Assume $\mu^\star = \mu_k^{1/2}(W)$. It is sufficient to show that with constant probability, we have $|B(S, \mu^\star)| \geq |W|/4$ (or $B(S, \mu^\star) \subseteq C$, equivalently). Let $\{C_i\}_{i \in [k]}$ be the optimal collection in the definition of $\mu_k^{1/2}(W)$, which means $\text{diam}(C_i) \leq \mu^\star$ for $i \in [k]$. Let $X_i$ be the indicator random variable for $S \cap C_i \neq \emptyset$. According to the uniform sampling from $W$, the probability that a point is sampled from $C_i$ equals $\alpha_i := |C_i|/|W|$. Hence, $\Pr[X_i = 1] = 1 - (1 - \alpha_i)^{|S|}$. If there is a point sampled from $C_i$, (i.e. $X_i = 1$), then all of the points in $C_i$ have a distance of at most $\mu^\star$ from the sampled point since $\text{diam}(C_i) \leq \mu^\star$. Hence,

$$\mathbb{E}[|B(S, \mu^\star)|] \geq \sum_{i=1}^{k} |C_i| \cdot \Pr[X_i = 1]$$
$$\geq |W| \cdot \sum_{i=1}^{k} \alpha_i \cdot (1 - (1 - \alpha_i)^{|S|}).$$

Note that $1 \geq \sum_{i=1}^{k} \alpha_i \geq 1/2$. The above function takes its minimum when $1/k \geq \alpha_1 = \alpha_2 = \cdots = \alpha_k \geq 1/(2k)$.[12] Thus,

$$\mathbb{E}[|B(S, \mu^\star)|] \geq |W| \cdot \sum_{i=1}^{k} \alpha_i \cdot (1 - (1 - \alpha_i)^{|S|})$$
$$\geq |W| \cdot (1/2) \cdot (1 - (1 - 1/(2k))^{|S|})$$
$$\geq |W| \cdot (1 - e^{-|S|/(2k)})/2 \geq \frac{1 - e^{-1}}{2} |W|.$$

Now, since we know that $0 \leq |B(S, \mu^\star)| \leq |W|$, we can conclude that with a constant probability we have $|B(S, \mu^\star)| \geq |W|/4$. More precisely, assume $p = \Pr[|B(S, \mu^\star)| \geq |W|/4]$, then we have

$$\frac{1 - 1/e}{2} |W| \leq \mathbb{E}[|B(S, \mu^\star)|] = \sum_{t=0}^{|W|} \Pr[|B(S, \mu^\star)| = t] \cdot t$$
$$\leq \sum_{t=0}^{|W|/4-1} \Pr[|B(S, \mu^\star)| = t] \cdot |W|/4$$
$$+ \sum_{t=|W|/4}^{|W|} \Pr[|B(S, \mu^\star)| = t] \cdot |W|$$

---

[12]This can be simply verified by elementary calculus.

$$= |W| \cdot ((1 - p)/4 + p)$$

This concludes $p \geq (1 - 2e^{-1})/3 = \Omega(1)$. $\qquad \square$

Now, we are ready to complete the proof of Lemma 3.5. Consider that $V^{\text{new}}$ is the current dynamic space, and let $x \in V^{\text{new}}$ be arbitrary. We show that $d(x, U) \leq 4 \cdot \text{OPT}_k(V^{\text{new}})$, where $U$ is the current output of the $\texttt{Sparsifier}$, which completes the proof. Let $i^\star \in [\ell]$ be the largest index such that $x \in U_{i^\star}$ (Since $U_1 = V^{\text{new}}$, this index exists). If $i^\star = \ell$, we obviously have $x \in U_\ell \subseteq U$ and $d(x, U) = 0$. Now, assume $i^\star < \ell$. We use the superscripts 'new' and 'old' to indicate the status of an object at the current time, and the last time that $\texttt{AlmostCover}$ was called on $U_{i^\star}$, respectively. For instance, the output of the last call $\texttt{AlmostCover}$ on $U_{i^\star}$ equals $(S_{i^\star}^{\text{old}}, U_{i^\star}^{\text{old}} \setminus C)$.

According to the definition of $i^\star$, We have that $x$ was not removed from the space between times old and new, as otherwise, $x$ became part of $U_{i^\star+1}$ and was not removed from $U_{i^\star+1}$ until the next call to $\texttt{AlmostCover}(U_{i^\star})$, which is a contradiction. We also have $x \in C$, as otherwise, $x$ would be inside $U_{i^\star+1}^{\text{new}}$. Hence, when we sampled points of $S_{i^\star}^{\text{old}}$, $x$ must have been assigned to the growing ball around some point $s \in S_{i^\star}^{\text{old}}$, which we denote by $B \subseteq C$. Note that during the updates between old and new, the center $s \in B$ might have been swapped with another point $s' \in B$ (Since $x$ was never removed from $B$, the ball $B$ never became empty and the algorithm maintains at least one point $s' \in S_{i^\star}^{\text{new}}$ as a representative for $B$). As a result, there exists $s' \in S_{i^\star}^{\text{new}}$ such that

$$d(x, s') \leq d(x, s) + d(s, s') \leq 2 \cdot \mu_k^{1/2}(U_{i^\star}^{\text{old}}). \quad (1)$$

The last inequality follows w.h.p. according to Claim 3.9 and the fact that we made $\Theta(\log n)$ independent calls to $\texttt{AlmostCover}(U_{i^\star}^{\text{old}})$ and select the best between all (see Lines 9 to 13 in Algorithm 1). Finally, we conclude

$$d(x, U) \leq d(x, S_{i^\star}^{\text{new}}) \leq d(x, s')$$
$$\leq 2 \cdot \mu_k^{1/2}(U_{i^\star}^{\text{old}}) \leq 2 \cdot \mu_k^1(U_{i^\star}^{\text{new}})$$
$$\leq 2 \cdot \mu_k^1(V^{\text{new}}) \leq 4 \cdot \text{OPT}_k(V^{\text{new}}).$$

The first inequality follows since $S_{i^\star}^{\text{new}} \subseteq U$, the second one follows since $s' \in S_{i^\star}^{\text{new}}$, the third one follows from Equation (1), the fourth one follow from Claim 3.8 for $W = U_{i^\star}^{\text{new}}$ and $W' = U_{i^\star}^{\text{old}}$ (note that $|U_{i^\star}^{\text{old}} \oplus U_{i^\star}^{\text{new}}| \leq \text{COUNT}[i^\star] \leq |U_{i^\star}^{\text{new}}|/4$), the fifth one follows from Claim 3.7, and the last one follows from Claim 3.6. As a result, w.h.p., we have $d(x, U) \leq 4 \cdot \text{OPT}_k(V^{\text{new}})$. $\qquad \square$

### 3.3.2. RECOURSE ANALYSIS

In this section, we prove the following lemma, which summarizes the recourse guarantee of the $\texttt{Sparsifier}$.

**Lemma 3.10.** *The amortized recourse of the algorithm* Sparsifier *is constant.*

The recourse caused by lazily updating the set $U$ after an insertion or deletion is at most $O(1)$ since only $O(1)$ many points are added or removed from each set. We now bound the recourse caused by periodically reconstructing the layers by using a *charging scheme*, similar to the one used by (Bhattacharya et al., 2023a) to bound the update time of this algorithm. Let $\sigma_1, \ldots, \sigma_T$ denote a sequence of updates handled by the algorithm. Each time the algorithm performs a reconstruction starting from layer $i$, we *charge* the recourse caused by this update to the updates that have occurred since the last time that $U_i$ was reconstructed. If the recourse caused by this update is $q$, and $p$ updates have occurred since the last time that $U_i$ was reconstructed, then each of these updates receives a charge of $q/p$. We denote the total charge assigned to the update $\sigma_t$ by $\Phi_t$. We can see that the amortized recourse of our algorithm is at most $(1/T) \cdot \sum_{t=1}^{T} \Phi_t$. We now show that, for each $t \in [T]$, $\Phi_t \leq O(1)$, implying that the amortized recourse is $O(1)$.

Consider some update $\sigma_t$ and let $U_1, \ldots, U_\ell$ denote the sets maintained by the algorithm during this update. The following claim shows that the sizes of the sets $U_i$ decay exponentially.

**Lemma 3.11** (Lemma B.1, (Bhattacharya et al., 2023a))**.** *There exists a constant $\epsilon \in (0,1)$ such that, for all $i \in [\ell]$, we have that $|U_\ell| \leq (1-\epsilon)^{\ell-i} \cdot |U_i|$.*

*Claim* 3.12. The recourse caused by the next reconstruction of layer $i$ is $O(k \log(|U_i|/k))$.

*Proof.* We first note that the size of a set $U_i$ changes by at most a constant factor before it is reconstructed. Since each $U_j$ has size $O(k)$, the recourse caused by reconstructing $U_i$ after the call to AlmostCover($U_i$) is at most $O(k(\ell - i + 1))$. Since the sizes of the sets decrease exponentially by Lemma 3.11, and $|U_\ell| = \Theta(k)$, it follows that $\ell - i = O(\log(|U_i|/k))$. Hence, the total recourse is $O(k \log(|U_i|/k))$. $\square$

Since the size of a set $U_i$ changes by at most a constant factor, we know that the recourse from reconstructing $U_i$ is charged to $\Omega(|U_i|)$ many updates. Hence, by Claim 3.12, the recourse charged to $\sigma_t$ from this reconstruction is $O(k \log(|U_i|/k)/|U_i|)$. Consequently, we can upper bound the total recourse charged to $\sigma_t$ by

$$\Phi_t \leq \sum_{i=1}^{\ell} O\left(\frac{k}{|U_i|} \log\left(\frac{|U_i|}{k}\right)\right).$$

The size of the set $U_\ell$ is always at least $9k$. This is because AlmostCover($U$) reduces the size of $U$ by a quarter, and combining with the condition in Line 15 in Algorithm 1, the

size of $U_\ell$ after each reconstruction is at least $16k \cdot (3/4) = 12k$. Now, during a sequence of at most $|U_\ell|/4$ updates before the next reconstruction from some layer $j \leq \ell$, the size of $U_\ell$ reduces to at most $12k \cdot (3/4) = 9k$ (even if all of the updates are deletions). Hence, we always have $|U_\ell| \geq 9k$. Now, by Lemma 3.11, it follows that $|U_i| \geq |U_\ell|/(1-\epsilon)^{\ell-i} \geq 9k/(1-\epsilon)^{\ell-i}$, so

$$
\begin{aligned}
\Phi_t &\leq O(1) \cdot \sum_{i=1}^{\ell} \frac{k}{|U_i|} \log\left(\frac{|U_i|}{k}\right) \\
&\leq O(1) \cdot \sum_{i=1}^{\ell} \frac{(1-\epsilon)^{\ell-i}}{9} \log\left(\frac{9}{(1-\epsilon)^{\ell-i}}\right) \\
&\leq O(1) \cdot \sum_{i=1}^{\ell} (1-\epsilon)^{\ell-i} (\ell - i + 1) \\
&\leq O(1) \cdot \sum_{j=0}^{\infty} (j+1)(1-\epsilon)^j \leq O(1),
\end{aligned}
$$

where the second inequality follows from the fact that $f(x) := \log(x)/x$ is decreasing for all $x \geq e$ and the last inequality follows from the fact that $\sum_{j=0}^{\infty}(j+1)(1-\epsilon)^j$ is a convergent series.

### 3.3.3. UPDATE TIME ANALYSIS

**Lemma 3.13.** *The amortized update time of the algorithm* Sparsifier *is $O(k \log^2 n)$.*

*Proof.* The subroutine AlmostCover($U$) can be implemented in $O(|S| \cdot |U|) = O(k \cdot |U|)$ time since we sample $|S| = O(k)$ centers, and compute the distance of any point $p \in U$ to $S$ in order to find $C$. Now, fix an index $i \in [\ell - 1]$. Each time that the algorithm reconstructs $U_{i+1}$, $\Theta(\log n)$ independent calls are made to AlmostCover($U_i$). Hence, the time consumed for reconstruction of $U_{i+1}$ is at most $O(k \cdot |U_i| \cdot \log n)$. Since the Sparsifier reconstructs $U_{i+1}$, whenever COUNT$[i] \geq |U_i|/2$ and then reset COUNT$[i]$ to zero, we conclude that the amortized running time of all calls to AlmostCover($U_i$) throughout the entire algorithm is $O(k \log n)$. Now, since $\ell = O(\log(n/k))$ (according to Lemma 3.4), the amortized update time of the Sparsifier is at most $O(\ell \cdot k \cdot \log n) = O(k \log^2 n)$. $\square$

## Conclusions and Future Work

We present a simple and almost optimal algorithm for dynamic $k$-center answering positively open questions in previous work. Natural future directions are improving the approximation factor, the recourse and the running time of our algorithm, or extending our results in the presence of outliers.

## Acknowledgments

Martín Costa is supported by a Google PhD Fellowship.

## Impact Statement

This paper presents work whose goal is to advance the field of Machine Learning. The nature of our work is mainly theoretical, and thus our insights might find use in various areas.

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

## A. State-of-the-Art for Dynamic $k$-Center

Table 1 contains a more comprehensive summary of the state-of-the-art algorithms for fully dynamic $k$-center in general metric spaces that we discuss in Section 1.

*Table 1.* State-of-the-art for **fully dynamic $k$-center** in general metrics. We distinguish between amortized and worst-case guarantees and specify whether guarantees hold in expectation or with high probability (when they do not hold deterministically). We also specify whether the randomized algorithms can deal with adaptive or oblivious adversaries.

| Reference | Approx. Ratio | Update Time | Recourse | Adversary |
|---|---|---|---|---|
| (Bateni et al., 2023) | $2 + \epsilon$ | $\tilde{O}(k/\epsilon)$ exp. amortized | $\Omega(k)$ | oblivious |
| (Lacki et al., 2024) | $O(1)$ | $\tilde{O}(\mathrm{poly}(n))$ worst-case | 4 worst-case | deterministic |
| (Forster & Skarlatos, 2025) | $O(1)$ | $\tilde{O}(\mathrm{poly}(n))$ worst-case | 2 worst-case | deterministic |
| (Bhattacharya et al., 2024a) | $O(\log n \log k)$ w.h.p. | $\tilde{O}(k)$ amortized | $\tilde{O}(1)$ amortized | adaptive |

## B. A Useful Lemma

**Lemma B.1.** *Given a metric space $(V, d)$, an integer $k \geq 1$ and $W \subseteq V$, we have that $\mathrm{OPT}_k(W) \leq 2 \cdot \mathrm{OPT}_k(V)$.*

*Proof.* Assume $U^\star \subseteq V$ is such that $\mathrm{cl}(U^\star, V) = \mathrm{OPT}_k(V)$. For each $u_i \in U^\star$, assume $s_i \in W$ is such that $d(u_i, s_i) = d(u_i, W)$. Let $S = \{s_1, s_2, \ldots, s_k\}$. Note that the size of $S$ might be less than $k$. $S$ is a feasible solution for the $k$-center problem on $W$. Fix an $x \in W$, and let $u_i \in U^\star$ is such that $d(x, u_i) = d(x, U^\star)$. We have

$$d(x, s_i) \leq d(x, u_i) + d(u_i, s_i) \leq 2 \cdot d(x, u_i) = 2 \cdot d(x, U^\star),$$

where the last inequality holds by $x \in W$ and the definition of $s_i$. As a result, for each $x \in W$, we have $d(x, S) \leq 2 \cdot d(x, U^\star)$. Finally,

$$\mathrm{OPT}_k(W) \leq \mathrm{cl}(S, W) = \max_{x \in W} d(x, S) \leq \max_{x \in W} 2 \cdot d(x, U^\star) \leq \max_{x \in V} 2 \cdot d(x, U^\star) = 2 \cdot \mathrm{OPT}_k(V). \quad \square$$

## C. Improving the Recourse to $8 + \epsilon$

In this section, we show how to use `Sparsifier` to obtain a new sparsifier called `BufferedSparsifier`, that can be combined with the algorithm `Dynamic-k-Center` to obtain a recourse of $8 + \epsilon$ for any arbitrary $0 < \epsilon \leq 1$.

### C.1. The Algorithm `BufferedSparsifier`

Define $q = 4/\epsilon$. We run the algorithm `Sparsifier` with the parameter $qk$, in order to maintain a space $W$ of size $\tilde{O}(qk)$ which is a 4-approximation for $\mathrm{OPT}_{qk}(V)$ with $\tilde{O}(qk)$ update time. Now, we show how to maintain the output $U$ of the `BufferedSparsifier` algorithm. First, we look at the current solution $W$ maintained by the `Sparsifier` on the current input space $V$, and let $U = W$ (and save all data structures inside the `Sparsifier` such as the centers in the $S_i$'s produced by internal calls to `AlmostCover` together with their clusters), which is a 4-approximation for the $(qk)$-center problem on $V$. Then, we perform $(q - 1)k$ *lazy* updates as follows. If a point $x$ is inserted into $V$, we also insert $x$ into $U$, and if a point $x$ is removed from $V$ and it is contained in $U$, we remove $x$ and add some arbitrary point in the cluster associated with $x$ to $U$ (if it is not empty).

After $(q - 1)k$ updates, we reset the set $U$, which means that we look at the current solution $W$ maintained by the `Sparsifier` and let $U^{\mathrm{new}} = W$. Now, we will remove all of the points that were previously in $U$ and insert all of the points in $U^{\mathrm{new}}$. These are considered as internal updates for `Dynamic-k-Center`, and during these updates, we do not

report the solution maintained by `Dynamic-k-Center`. We report the solution only after we have removed points of the previous $U$ and inserted $U^{\text{new}}$. Hence, the solution maintained by `Dynamic-k-Center` can change completely, but incur a recourse of at most $2k$ (the previous solution was a subset of size $k$ in $U$ and the new one is a subset of size $k$ in $U^{\text{new}}$). Now, starting from this new set $U^{\text{new}}$, we continue with lazy updates for $(q-1)k$ many updates as described before, and this process continues.

The difference between `BufferedSparsifier` and `Sparsifier` is that, although the recourse of `Sparsifier` can be a large constant, the amortized recourse of `BufferedSparsifier` is small. The reason is that we copy the solution $W$ maintained by `Sparsifier` and then perform lazy updates that each lead to low recourse, for a long period of time ($(q-1)k$ updates). Since the copied space $W$ is a good approximation of $\text{OPT}_{qk}(V)$, we show that it remains a good approximation for $\text{OPT}_k(V)$ during this longer period.

### C.2. Analysis of `BufferedSparsifier`

In order to analyze the approximation guarantee of `BufferedSparsifier`, we need the following lemma, known as the Lazy-Updates Lemma.

**Lemma C.1** (Lemma 3.3 in (Bhattacharya et al., 2024a))**.** *Assume $V$ and $V'$ are two subsets of a ground metric space $\mathcal{V}$ such that $|V \oplus V'| \leq s$ for some $s \geq 0$. Then, for every $k \geq 1$, we have $\text{OPT}_{k+s}(V) \leq \text{OPT}_k(V')$.*

**Lemma C.2.** *The approximation ratio of `BufferedSparsifier` is 4.*

*Proof.* Consider a point in time where we set $U = W$ (the solution maintained by the `Sparsifier`). Assume $V^0$ is the current input space and $U^0 = U$. It is sufficient to show that the maintained solution $U$ is always a 4 approximation for the $k$-center problem on the current space $V$ after $i$ updates for any arbitrary $0 \leq i \leq (q-1)k$. Fix $i$ and let $U'$ and $V'$ be the maintained solution and the input metric space after $i$ updates, respectively. Let $I := V' \setminus V^0$ be the set of inserted points to $V$. Assume $D := U^0 \setminus U'$ is the set of deleted points from $V$ that affect $U$, and are replaced with arbitrary points in their clusters. Hence, $U' = U^0 - D + R + I$, where $R$ is the set of centers that are replaced with centers in $D$.

According to the procedure of the `BufferedSparsifier`, since every point in $I$ is inserted into $U$, they do not increase the cost of the solution $U$, i.e. $\text{cl}(U', V') \leq \text{cl}(U^0 - D + R, V^0)$. According to the 4-approximation analysis for `Sparsifier` in Section 3.3.1, whenever a point $u \in D$ is replaced with any arbitrary point within its cluster, the solution remains a 4-approximation (see the discussion at the end of Section 3.3.1). Hence, $\text{cl}(U^0 - D + R, V^0) \leq 4 \cdot \text{OPT}_{qk}(V^0)$. Combining the two previous inequalities with Lemma C.1, for every $0 \leq i \leq (q-1)k$, we have that

$$\text{cl}(U', V') \leq \text{cl}(U^0 - D + R, V^0) \leq 4 \cdot \text{OPT}_{qk}(V^0) \leq 4 \cdot \text{OPT}_{qk-i}(V') \leq 4 \cdot \text{OPT}_k(V').$$

The last inequality follows since $qk - i \geq k$. As a result, during the lazy updates, the bicriteria solution $U$ is always a 4-approximation for the $k$-center problem on the original metric space. $\qquad\square$

**Lemma C.3.** *The amortized recourse of the algorithm obtained by composing `Dynamic-k-Center` with the sparsifier `BufferedSparsifier` is at most $8 + \epsilon$.*

*Proof.* During the lazy updates, for every insertion in $V$, we have an insertion in $U$, and for every deletion from $V$, we have at most two updates in $U$ (deleting a center and replacing it with any arbitrary point within its cluster). Hence, the length of the input stream (updates on $U$) which is fed into `Dynamic-k-Center` is at most twice the length of the main input stream. We incur a recourse of at most 4 in expectation w.r.t. the input stream fed to `Dynamic-k-Center` according to the recourse analysis of the algorithm in Section 2.3, which concludes that the total amortized recourse w.r.t. the main input stream is at most 8. After restarting $U$, we incur a recourse of at most $2k$ since we have two solutions of size $k$. Note that the internal changes in the main solution during deletion of points in the previous $U$ and adding the points in the new $U$ does not count as recourse for the final algorithm, since we only report the final solution after all of these updates as the solution by the algorithm maintained.

Since the above procedure happens only every $(q-1)k$ updates, the total amortized recourse incurred by these updates is at most $2/(q-1)$. As a result, the final recourse of the algorithm is $8 + 2/(q-1) \leq 8 + \epsilon$. $\qquad\square$

*Remark* C.4. In many real-world applications of the $k$-center problem, the dynamic input space is always a subset of a static ground metric space (such as $\mathbb{R}^n$ in the Euclidean $k$-center problem), and it is allowed to open any arbitrary point from the

ground metric space as a center. It is possible to tweak `BufferedSparsifier` in order to provide a sparsifier which is not necessarily a subset of the current dynamic space $V$, but has the following guarantees. By feeding the output of this sparsifier into `Dynamic-$k$-Center`, we still have a 20 approximation algorithm. The amortized recourse of the final algorithm becomes $4 + \epsilon$ instead of $8 + \epsilon$. The tweak is simple and works as follows. Whenever a point $x$ is removed from the space and it is contained in a $S_i$, instead of replacing $x$ with any arbitrary point in the cluster of $x$, we just keep $x$ as it is. The only time that we remove $x$, is whenever the cluster of $x$ becomes empty (all of the points in the cluster of $x$, including $x$ itself, are removed from the original space). With a similar argument, it is possible to show that any 8-approximate solution w.r.t. $U$ is a 20-approximation solution w.r.t. the original space. The recourse of $4 + \epsilon$ also follows immediately with the previous arguments. As a result, if we are allowed to open centers from outside the current dynamic space $V$, it is possible to achieve a dynamic $k$-center algorithm with an approximation ratio of 20 and a recourse of $4 + \epsilon$.

