# OpenReview forum: "Almost Optimal Fully Dynamic $k$-Center Clustering with Recourse"
_ICML.cc/2025/Conference — ICML 2025 poster_

### Official Review · Reviewer_WqgH · 2025-03-09

**Overall Recommendation:** 3

**Summary:**

The paper claims to construct an $O(1)$-approximate solution for the metric $k$-center problem in the dynamic setting, with $O(1)$ amortized recourse and $\widetilde{O}(k)$ amortized update time. By combining a recursively nested MIS (Maximal Independent Set) with a dynamic sparsifier, the paper improves the amortized update time of MIS from $\widetilde{O}(n)$ to $\widetilde{O}(k)$.

**Claims And Evidence:**

The proofs for MIS are clear and convincing. However, the explanation of the dynamic sparsifier lacks clarity; more details or a refined lemma for the properties of the dynamic sparsifier may be useful.

**Essential References Not Discussed:**

No

**Ethics Expertise Needed:**

["Other expertise"]

**Experimental Designs Or Analyses:**

No experiments are included.

**Methods And Evaluation Criteria:**

The methods are appropriate. The paper adapts the dynamic sparsifier from $k$-means to $k$-center via a conversion from $k$-means to $(k,p)$-clustering and then to $k$-center.

**Other Comments Or Suggestions:**

No

**Other Strengths And Weaknesses:**

The work demonstrates originality by extending the dynamic sparsifier from $k$-means to $k$-center. The notation and lemmas are clearly presented, and the proof ideas are straightforward.

**Questions For Authors:**

Mentioned above

**Relation To Broader Scientific Literature:**

The paper claims to achieve optimal approximation ratio, amortized recourse, and update time simultaneously, building on prior work.

**Theoretical Claims:**

The proofs and analysis for MIS are correct. However, the analysis of amortized recourse and update time in the dynamic sparsifier is unclear and raises questions:
1) In the expression "$\beta · k = \wiletilde{O}(k)$," is a $\log n$ factor hidden?
2) On page 8, the paper states that "each $U_j$ has size $O(k)$." However, since $U_1 = O(n)$, could $U_j$ potentially be $O(n)$? If so, should the recourse here include a $\log n$ term?

---

> ### Author Rebuttal · Authors · 2025-03-31
>
> We thank all the reviewers for their efforts and insightful comments. Please let us know if you have any questions about our rebuttal.
>
> Thank you for pointing this out; this will help us improve the clarity of the proofs.
> Indeed, since $\beta = O(\log (n/k))$,
> there is a $\log (n/k)$ factor hidden in  $\Tilde{O}(k) = O(k \cdot \log (n/k))$, which also adhere our use of notation $\Tilde{O}$ (please see footnote $2$ in page $2$).
> We will point this out clearly in the Notation section instead of the footnote.
> Note that this will not break down the proof of Theorem 1.3 since no matter what is the size of the space (either $n$ before using sparsifier, or $\beta \cdot k = k \log (n/k)$ after using sparsifier) the recourse would be constant since we are calling the algorithm of Theorem 1.2 on this space which has constant recourse (i.e. $R_A(\cdot) = O(1)$ in Line 312).
>
> You are right about the comment on page 8. There is a typo in Line 435, we should have written: ``each $S_j$ has size $O(k)$".
> The rest of the analysis remains intact and correct.
> Due to space constraints, we could not elaborate more on the proofs in the paper.
> Below, we explain the proof in more detail (note that the current proof in the paper is self-contained and this explanation does not add any new ideas or change the proof):
>
> Clarification on the proof:
> Note that the recourse is defined as the number of changes in the output of the sparsifier $U = S_1 \cup S_2 \cup \cdots \cup S_{\ell-1} \cup U_\ell$ (see line 13 of Algorithm 1), and by reconstructing from $U_i$, the sets undergoing changes in the output are $S_i, S_{i+1}, \ldots, S_{\ell-1}$ and $U_\ell$.
> This concludes the total change in the output $U$ is at most $O(k(\ell-i+1))$ since each $S_j$ has size $O(k)$, as well as $U_\ell$.
> Although $|U_1| = O(n)$, the sizes of $U_i$s decrease \textbf{exponentially} according to Lemma 3.11, which is the key fact that we use to bound the recourse.
>
> The recourse caused by reconstructing from $U_i$ is at most $O(k(\ell-i+1))$ as explained above.
> Since the sizes of $U_i$s decrease exponentially (according to Lemma 3.11) until it becomes $|U_\ell| = \Theta(k)$, we get $\ell - i = O(\log(|U_i|/k))$.
> Since we only start the reconstruction from $U_i$ every $\Omega(|U_i|)$ steps, the amortized recourse incurred by all the reconstructions starting from level $i$ would be bounded by $\frac{k}{|U_i|} \cdot \log \left( \frac{|U_i|}{k} \right)$.
> The rest of the analysis follows from the bound on the potential function defined in the paper.

---

> > ### Comment · Reviewer_WqgH · 2025-04-02
> >
> > Thanks for the response. I have one more question: Given that there is $\mathrm{poly} \log n$ factor in the amortized update time, does your result ultimately improve any result in Table 1, e.g., Bateni et al., 2023, or is the new result parallel to them? Could you provide a more straightforward comparison between your results and those of the previous ones, e.g., listing the advantages and disadvantages of the new result?

---

> > > ### Author Response · Authors · 2025-04-03
> > >
> > > Thank you for your response.
> > > First, we note that although the update time of the algorithm is important, the approximation ratio and the recourse are also two crucial parameters to optimize in $k$-clustering problems.
> > > The notion of recourse in $k$-clustering has received significant attention in recent years.
> > > We now elaborate on the advantages of our algorithm:
> > >
> > > We did not optimize the analysis of the constants in our paper for the sake of simplicity.
> > > We will provide a more intricate analysis in the updated version of our paper that shows the approximation ratio is at most $20$---this is the best approximation ratio of any known algorithm with constant recourse.
> > > In particular, the previous smallest approximation was $24$ by Lacki et al.~[SODA24], but their update time is a large $O(\text{poly} (n))$.
> > >
> > > Furthermore, by slightly tweaking how we use the sparsifier, we can get a recourse of at most $4 + \epsilon$, for any constant $\epsilon > 0$. We note that the best we can hope to achieve here is a recourse of $2$, since in the worst case, we must remove a point and insert a new point.
> > > We note that the result of Bateni et al.~[SODA23] does not have any guarantee on recourse (it might be as large as $\Omega(k)$).
> > > We will provide this analysis in the updated version of the paper.
> > >
> > > Finally, the update time of our algorithm is some $\text{poly}\log n$ factor larger than Bateni et al.~[SODA23].
> > >
> > > To summarize, our approximation ratio is better than all of the previous results that have constant recourse, and the update time is at most some $\text{poly}\log n$ factor larger than Bateni et al.~[SODA23], which does not have any guarantee on recourse.
> > >
> > > We will make sure to add a more detailed comparison in our updated paper.

---

### Official Review · Reviewer_342V · 2025-03-11

**Overall Recommendation:** 4

**Summary:**

This paper gives almost optimal dynamic $k$-center algorithm in metric space. In dynamic $k$-center, the task is to obtain an approximate solution with as small update time and recourse, where recourse means the number of center points needed to be updated per insertion/deletion.

This paper designs an algorithm to obtain $O(1)$-approximation with $\tilde{O}(k)$ update time and $O(1)$ recourse. So it achieves improved or matching guarantees in all metrics (approximation factor, update time, recourse) upon all existing results. This is impressive.

Technically, authors borrow a few techniques from the literature and manage to combine all the advantages.

**Claims And Evidence:**

The claims have been proved rigorously.

**Essential References Not Discussed:**

No

**Experimental Designs Or Analyses:**

Not applicable.

**Methods And Evaluation Criteria:**

Yes

**Other Comments Or Suggestions:**

I have not found typos. The writing is very good.

**Other Strengths And Weaknesses:**

Strength: 1. The algorithm presented is easy to follow. 2. The paper is mostly written well.

Weakness: No experiments.

**Questions For Authors:**

1. Are there any evidences (e.g. a lower bound) that $\tilde{O}(k)$ is a necessary update time bound?
2. Which metric of the algorithm performance suffers failure probability? I have not found the explicit probability claim in Theorem 1.1. Is it the case that with high probability the algorithm succeeds at every update?

**Relation To Broader Scientific Literature:**

Dynamic k-center is an important research problem in algorihtmic machine learning. This paper achieves state-of-the-art dynamic k-center algorithm. I believe the result will benefit future research.

**Theoretical Claims:**

I read the proofs and I think they are correct.

---

> ### Author Rebuttal · Authors · 2025-03-31
>
> We thank all the reviewers for their efforts and insightful comments. Please let us know if you have any questions about our rebuttal.
>
> It is known that an update time of $\Omega(k)$ is necessary for this problem.
> As stated in Line 58 in the introduction, it is known that, in the static setting, a running time of $\Omega(nk)$ is necessary to achieve any non-trivial approximation (Bateni et al. 2023).
> Any dynamic algorithm with $o(k)$ update time yields a static algorithm with $o(nk)$ running time, which rules out any non-trivial approximation ratio.
> Hence, $\Omega(k)$ update time is necessary in the dynamic setting, which means that $\tilde{O}(k)$ update time is near optimal (up to polylog factors).
>
> Indeed, when we say that our algorithm succeeds with high probability at every update, we mean that the approximation ratio holds with high probability after handling each update (please see Footnote 5 in Theorem 1.3).

---

### Official Review · Reviewer_hRcL · 2025-03-11

**Overall Recommendation:** 3

**Summary:**

The paper proposes an algorithm for the k-center problem in the fully-dynamic setting in general metric spaces. In particular, the proposed method obtains a constant approximation with constant recourse and Otilde(k) update time (thus the name “almost optimal”). The algorithm is based on a combination of the reduction of the k-center problem to dynamic MIS in threshold graphs by Bateni et al. with the dynamic sparsifier (for k-median) of Bhattacharya et al.

**Claims And Evidence:**

The theoretical claims seem to be correct. However, they are often very imprecise (see W1 and W2).

**Essential References Not Discussed:**

Some additional references (albeit not essential) could be discussed, see W3.

**Experimental Designs Or Analyses:**

Not applicable.

**Methods And Evaluation Criteria:**

The proposed method, which combines two existing theoretical tools, is simple and very effective.

**Other Comments Or Suggestions:**

Minor comments:
- You state that the fully-dynamic framework focuses mainly on the three metrics, approximation ratio, recourse and update time. This is a compelling story for your paper. However, I would argue that there is a fourth metric, the query time. This is arguably the most important one. Saving all the points in a set and re-running Gonzalez on the pointset after each update has very low update time and yields the optimal approximation guarantee, but of course it has terrible query time, and thus does not qualify as a fully dynamic algorithm. This should be mentioned at least briefly.


Remark: Despite the simplicity of the proposed method, I am prone to suggesting acceptance if the authors address (e.g. by stating clearly the modifications they would make to the manuscript) my concerns on (i) the clarity in the statements of the theorems, and also on (ii) the literature review and (iii) the clarity of proofs.

Edit: after the rebuttal, I increased my score to suggest acceptance.

**Other Strengths And Weaknesses:**

Strengths:
- S1: the k-center problem is a highly relevant topic, and the fully dynamic framework is of great interest to both theoreticians and practitioners.
- S2: the paper is technically very solid. the contributions, although incremental on the works on which is based, are non-trivial and seem to be correct. Moreover, the paper represents a step forward in closing the gap towards an optimal algorithm for dynamic k-center.

---

Weaknesses:
- W1 (Main concern): Hiding the logarithmic factors in the $\widetilde{O}$ notation is fine for the statements in the introduction. However, in the rest of the paper, the correct asymptotic running time should be stated. Indeed, the current notation would make it harder for subsequent works to compare the running times to the ones of the present work (e.g. the comparison for reasonable values of $k$ between a $O(k \sqrt k)$ algorithm and a $O(k  (\log k)^{10} )$ algorithm would be misrepresented by the $\widetilde{O}$ notation.)
The same actually holds for the approximation ratio and recourse, they should be clearly stated in the theorem statements in the technical sections. I would say this is especially important in a venue like ICML, which is of interest also to practitioners (unlike STOC or FOCS).
- W2: the fact that the algorithm assumes oblivious adversaries, although fair, should be stated in Theorem 1.1, as not to overstate results. It is unclear to me why the authors chose to duplicate the theorem into Thm 1.1 and 1.3.
- W3: The discussion of related works seems insufficient, as it fails to discuss related works such as:
    * [1] improves memory requirements over the original fully dynamic k-center paper, at the cost of a worse approx. ratio.
    * [2] 2+eps in general metric spaces, efficient in spaces with low doubling dimension.
    * [3] 2+eps in general metric spaces, state-of-the-art for update times in spaces with low doubling dimension. Also, k and eps are not needed to be known in advance, which offers an advantage over the present work.
    * [4] 2+eps approximation, and offers an alternative approach for spaces with low doubling dimension. Also does not require k and eps to be known.
    * [5] proposes an algorithm for dynamic k-center on graphs.

These works and the relationship to the present work should be discussed in the related work section or in Table 1. See for example Table 1 in [4].

- W4: The proofs, albeit correct, are not easy to follow, as they require jumping back and forth in the paper. As an example, the recourse analysis in Thm 1.3 is proved for the sparsifier in Section 3.4.2, but then requires the reader to backtrack to Lemma 3.3 to recall how this is used to obtain the total recourse. A few sentences here and there to guide the reader would go a long way.
- W5: As previously stated, the paper is incremental with respect to the works on which is based. Indeed, other than combining the two established techniques, it seems like the recourse analysis of the sparsifier is the only substantial technical contribution.

---
- [1] Chan, T-H. Hubert, et al. "Fully Dynamic k-Center Clustering With Improved Memory Efficiency." IEEE Transactions on Knowledge and Data Engineering (2020).
- [2] Goranci, Gramoz, et al. "Fully dynamic k-center clustering in low dimensional metrics." 2021 Proceedings of the Workshop on Algorithm Engineering and Experiments (ALENEX).
- [3] Pellizzoni, Paolo, Andrea Pietracaprina, and Geppino Pucci. "Fully dynamic clustering and diversity maximization in doubling metrics." Algorithms and Data Structures Symposium. 2023.
- [4] Gan, Jinxiang, and Mordecai J. Golin. "Fully dynamic k-center in low dimensions via approximate furthest neighbors." Symposium on Simplicity in Algorithms (SOSA) 2024
- [5] Cruciani, Emilio, et al. "Dynamic algorithms for 𝑘-center on graphs." arXiv preprint arXiv:2307.15557 (2023).

**Questions For Authors:**

-

**Relation To Broader Scientific Literature:**

The paper combines two existing theoretical tools from the literature in a non-trivial way. This paper represents a step forward in closing the gap towards an optimal algorithm for dynamic k-center.

**Theoretical Claims:**

I checked the proofs, albeit not in depth, and they seem to be correct. However, they often lack clarity.

---

> ### Author Rebuttal · Authors · 2025-03-31
>
> We thank all the reviewers for their efforts and insightful comments. Please let us know if you have any questions about our rebuttal.
>
> Our algorithm can be implemented to have an update time of $O(k \cdot \log^4 n \log \Delta)$ by using standard data structures.
> Regarding the approximation ratio and recourse, we did not optimize our analysis of the constants in these bounds for the sake of simplicity.
> By carrying out a more intricate analysis of the sparsifier, we can show that our approximation ratio is at most $20$---this is the best approximation ratio of any algorithm with constant recourse, even if we allow exponential update time. In particular, the previous smallest was $24$ by Lacki et al. [SODA'24].
> Furthermore, by slightly tweaking how we use the sparsifier, we can get a recourse of at most $4 + \epsilon$, for any constant $\epsilon > 0$ (we note that the best we can hope to achieve here is a recourse of $2$, since in the worst case we must remove a point and insert a new point).
> We tried to keep the proofs simple and provide asymptotic bounds.
> Although the constants in the approximation ratio and recourse are not specified in the current analysis, they are good in practice.
>
> We implemented our algorithm and compared the approximation ratio of our algorithm with the static $2$-approximation offline greedy algorithm by Gonzalez; we observed that the quality of the solution maintained by our algorithm is significantly better than what is derived by the current analysis.
> Specifically, we tested our dynamic algorithm on 5 different datasets, on inputs of size $n = 10000$ with $k = 10, 50$ and $100$, and compared the cost of the solution maintained by our algorithm to the cost of the solution produced by the offline greedy algorithm. We observed that the cost of the solution produced by our algorithm is consistently within a factor of 1.25 - 1.75 of the cost of the greedy algorithm.
>
> Regarding recourse, we also observed that the amortized recourse of our algorithm is actually sublinear in practice, at most $1$ in almost all of our test results.
>
> We will improve the accuracy of the statements in the paper and provide precise guarantees.
> We will also improve the analysis and provide a clear road map together with a natural flow for the analysis to be easy to follow for the reader.
>
> Thank you for pointing out the results that we missed mentioning in our paper.
> We will add them to the introduction and the related work sections of our paper.
>
> Regarding the query time, note that our algorithm maintains the solution explicitly after every update.
> Without considering an explicit solution after every update, we can not define the notion of recourse.
> In this framework, the query time is subsumed by the update time since we can assume there is a query after every update as the solution must be maintained explicitly.
> We will mention this clearly in the introduction to prevent any confusion.

---

> > ### Comment · Reviewer_hRcL · 2025-04-02
> >
> > Thank you for your reply.
> > I think you should include the time complexity explicitly, as it would be interesting to try to close the gap with $O(k \log\Delta)$ or even $O(k)$ (but it's unlikely). As for the approximation ratio, the fact that it is tight with the best possible one (for constant-recourse algorithms) should be highlighted. If the analysis deviates significantly from the current one, you might want to defer the formal proof to an extended version of the paper, and only state it as a high-level remark. For the recourse, I'd advise against modifying the strategy, as the reviewers would have no way of assessing it. An analysis of the current one would be sufficient.
> >
> > I am not surprised that the algorithm behaves in practice better than the worst case analysis suggests, and that's also a positive point.
> >
> > I am confident that the revised paper with the more accurate statements, the expanded related work and a clear roadmap would be a significant improvement over the current state. Therefore, I am willing to increase my score.

---

> > > ### Author Response · Authors · 2025-04-03
> > >
> > > Thanks for your insightful suggestions. We will follow your recommendations to make our paper stronger.

---

### Official Review · Reviewer_GqKD · 2025-03-13

**Overall Recommendation:** 4

**Summary:**

There is no better summary of the paper than the one given in the abstract of the paper. So, I'll simply copy it below:
_"We give a simple algorithm for dynamic k-center that maintains an O(1)-approximate solution with O(1) amortized recourse and ˜O(k) amortized update time, obtaining near-optimal approximation, recourse and update time simultaneously. We obtain our result by combining a variant of the dynamic k-center algorithm of Bateni et al. [SODA’23] with the dynamic sparsifier of Bhattacharya et al. [NeurIPS’23]."_

Let me expand on some of the terms used to better understand the above statement.
- Dynamic algorithm: The algorithm maintains a solution for the problem at every step of data update, which includes both insert and delete operations on data items.
- The time to update the solution at every step is the update time, and the number of changes to the maintained solution is the recourse. The amortised analysis considers the overall resource usage across all steps instead of considering the worst-case over all the steps.

-The k-center problem is NP-hard to approximate below 2-approximation. Given this, O(1) approximation is the best one can hope for.
- The minimum number of steps required to obtain any constant approximation for k-center is \Omega(nk). Given this, O(k) update time is the best one can hope for.
So, the result is nearly optimal in approximation factor, update time, and recourse. Previous results did not achieve these almost-optimal bounds on ALL of (approximation, update, recourse).

**Claims And Evidence:**

Yes, the claims made in the paper are supported by evidence.

**Essential References Not Discussed:**

I found the references are appropriate.

**Experimental Designs Or Analyses:**

This is a theoretical paper. There are no experiments.

**Methods And Evaluation Criteria:**

Yes, the evaluation criteria makes sense.

**Other Comments Or Suggestions:**

- Some comments have been mentioned in the strengths and weaknesses section.

## Update after rebuttal:
I have decided to retain my score of (4:accept) after the rebuttal.

**Other Strengths And Weaknesses:**

Strengths:
- The paper achieves almost optimal bounds for all the relevant resource bounds in a dynamic algorithm for the k-center problem (an important clustering problem).
- This is an important progress from the theoretical viewpoint. The paper brings together several ideas from dynamic algorithms literature and utilises them to design a state-of-the-art dynamic algorithm.
- The high-level ideas in the paper are well presented.



Weaknesses:
- It would be good to see experimental results that compare the given algorithm with other known dynamic algorithms.
- Instead of stating O(1) approximation, it might be good to explicitly state the constant in the constant factor approximation (unless the constant is bad). If this constant equals 2, then a brief discussion on where the loss happens could help close the gap.
- Briefly describing DynamicMIS used in Lemma 2.1 should help the reader unfamiliar with the previous literature. For example, it seems from the statement of Lemma 2.1 that DynamicMIS is a randomised algorithm (hence the expectation). So, it should be mentioned clearly that the expectation is over the internal randomness of the algorithm. This also makes is clear that the proposed algorithm is a randomised algorithm.

**Questions For Authors:**

- Some important questions have been mentioned in the strengths and weaknesses section.

**Relation To Broader Scientific Literature:**

The paper makes non-trivial progress on a theoretical problem on dynamic k-center problem. This is an important addition to the theory of clustering.

**Theoretical Claims:**

I verified the claims at a high level, and the claims are sound. It is possible that I may have missed some specific details.

---

> ### Author Rebuttal · Authors · 2025-03-31
>
> We thank all the reviewers for their efforts and insightful comments. Please let us know if you have any questions about our rebuttal.
>
> In order to keep the proofs simple, we did not optimize the constant in the approximation ratio in the paper. By carrying out a more intricate analysis of the sparsifier, we can show that our approximation ratio is at most $20$---this is the best approximation ratio of any algorithm with constant recourse, \emph{even if we allow exponential update time}. In particular, the previous smallest was $24$ by Lacki et al. [SODA'24].
> We have also done some experiments and compared the approximation ratio of our algorithm with the static $2$-approximation offline greedy algorithm by Gonzalez; we observed that the quality of the solution maintained by our algorithm is significantly better than what is derived by the current analysis.
> Specifically, we tested our dynamic algorithm on 5 different datasets, on inputs of size $n = 10000$ with $k = 10, 50$ and $100$, and compared the cost of the solution maintained by our algorithm to the cost of the solution produced by the offline greedy algorithm. We observed that the cost of the solution produced by our algorithm is consistently within a factor of 1.25 - 1.75 of the cost of the greedy algorithm.
>
> Thank you for pointing out the issue with using the DynamicMIS algorithm as a black box.
> We will elaborate on this and clarify this point.

---

### Decision · Program_Chairs · 2025-05-01

**Decision:**

Accept (poster)

**Comment:**

The paper presents a simple algorithm for the dynamic $k$-center problem that achieves a constant-factor approximation, $O(1)$ amortized recourse, and $\tilde{O}(k)$ amortized update time, matching near-optimal bounds across all three measures. Although the results are qualitatively meaningful, the two main initial areas of concern stemmed around
- Imprecise guarantees for approximation, hidden in $O(1)$ notation
- Lack of experiments

As the focus of the paper is theoretical, the lack of experiments is not crucial. Moreover, the author response clarified the precise terms in the $O(1)$ notation, thereby resolving the initial reviewer concerns, provided these changes are reflected in future versions.